# Improvement of Post-Operative Quality of Life in Patients 2 Years after Minimally Invasive Surgery for Pain and Deep Infiltrating Endometriosis

**DOI:** 10.3390/jcm11206132

**Published:** 2022-10-18

**Authors:** Sophie Legendri, Marie Carbonnel, Anis Feki, Gaby Moawad, Gabrielle Aubry, Alexandre Vallée, Jean-Marc Ayoubi

**Affiliations:** 1Department of Obstetrics and Gynecology, Foch Hospital, 40 Rue Worth, 92150 Suresnes, France; 2Medical School, University of Versailles, Saint-Quentin-en-Yvelines, 55 Avenue de Paris, 78000 Versailles, France; 3Department of Obstetrics and Gynecology, Cantonal Hospital Fribourg, 1702 Fribourg, Switzerland; 4Department of Obstetrics and Gynecology, The George Washington University Hospital, Washington, DC 20037, USA; 5Department of Clinical Research and Innovation, Foch Hospital, 40 Rue Worth, 92150 Suresnes, France

**Keywords:** endometriosis, surgery, robotic-laparoscopy, laparoscopy, quality of life, EPH-5, total surgery, conservative surgery

## Abstract

This study addressed the improvement in the quality of life of patients 2 years after minimally invasive surgery for painful deep infiltrating endometriosis (DIE), evaluated with EHP-5 (Endometriosis Health Profile-5) scores and the intensity of dysmenorrhea and dyspareunia. This was a retrospective study, performed in a referral centre for endometriosis, between January 2010 and January 2019. EHP-5 scores were complete for 54 patients, and two subgroups were analysed: classic laparoscopy (CL) vs. robotic laparoscopy (RL), and conservative surgery (ConservS) vs. total surgery (TS). There was an important decrease in 2-year post-operative EHP-5 scores in the global population (pre-op: 61.36 (42.18–68.75) and 2-year post-op: 20.45 (0–38.06); *p* < 0.001). The Visual Analogic Scale (VAS) was also lower for dysmenorrhea (pre-op: 8 (7–9.75) vs. 2-year post-op: 3 (2–5.25); *p* < 0.001) and dyspareunia (pre-op: 6 (3.1–8.9) vs. 2-year post-op: 3 (0–6); *p* < 0.001). In the subgroup analysis, EHP-5 scores were improved in the RL group (pre-op: 65.9 (59.09–71.02) vs. 2-year post-op: 11.4 (0–38.06); *p* < 0.001) and the CL group (pre-op: 50 (34.65–68.18) vs. 2-year post-op: 27.27 (14.20–40.90); *p* < 0.001), with a slight advantage for RL (*p* = 0.04), and the same improvements were found for ConservS (pre-op: 61.4 (38.06–71.59) vs. 2-year post-op: 22.7 (11.93–38.07); *p* < 0.001) and TS groups (pre-op: 61.59 (51.70–68.75) vs. 2-year post-op: 13.63 (0–44.30); *p* < 0.001). Minimally invasive surgery improved the quality of life for DIE patients 2 years after surgery, and conservative surgery showed comparable results to total surgery.

## 1. Introduction

Endometriosis is a complex multifactorial disease that affects 2–10% of patients of childbearing age [1]. Deep infiltrating endometriosis (DIE) may cause chronic pelvic pain, dysmenorrhea, dyspareunia, and infertility, and may result in impaired quality of life [2,3]. Pain management is based on a holistic approach. If surgery is necessary, recommendations for symptomatic deep endometriosis favour extensive surgery to excise all endometriosis lesions and reduce the risk of recurrence [1,4,5], which is estimated at 2–12% [6,7]. This surgery has demonstrated a low recurrence rate but a higher risk of long-term complications, especially in cases of colorectal or urinary tract surgery [8,9], and has a significant effect on quality of life [10]. Minimally invasive surgery using laparoscopy is the preferred approach, when feasible, for the management of painful DIE [4]. Robotic laparoscopy (RL) has proven its feasibility in DIE, permitting better dexterity and having a shorter learning curve in comparison with conventional laparoscopy [11,12]. There are no differences to date in terms of complications, bleeding, or efficiency between RL and classic laparoscopy (CL), apart from the duration of surgery, which tends to be longer in RL [13]. However, only a few small studies are available [13]. No team has evaluated the improvements in quality of life when comparing the two surgical routes, or total vs. conservative excision of DIE. 

The principal aim of this study was the improvement in the quality of life when using EHP-5 [14], before and 2 years after minimally invasive surgery for painful DIE. We analysed subgroups comparing classic laparoscopy (CL) vs. robotic laparoscopy (RL), and conservative surgery (ConservS) vs. total surgery (TS).

## 2. Materials and Methods

### 2.1. Patients and Data Collection

We conducted a retrospective study on a database spanning 10 years in a single French referral centre for endometriosis at Foch Hospital. A search was performed for specific keywords such as “laparoscopy”, “robotic laparoscopy”, “deep infiltrating endometriosis”, “endometriosis”, “surgical treatment”, “total surgery”, and “conservative surgery” from January 2010 to January 2019. Eligible patients had minimally invasive surgery for painful DIE after failure of medical management. The exclusion criteria were all cases of surgery performed for any other main reason than pain (infertility, menorrhagia, adenomyosis), superficial endometriosis, non-symptomatic endometriomas, and extra-peritoneal endometriosis without associated DIE. Patients who followed a laparotomy route (5 patients: 1 complex vesical resection and 4 digestive resections with anastomosis) and patients who did not fill out the EHP-5 questionnaire were also excluded.

DIE was characterised in this study as the involvement of endometrial-like tissue at a depth of more than 5 mm [4]. All patients had AFSR (revised American Fertility Society) scores > 40 points (stage 4). They underwent preoperative MRI and pelvic US to map the disease. No case of misdiagnosis, especially malignant tumour, was revealed in histopathology analysis. The CL and RL interventions were performed by four experienced surgeons. The types of surgeries were collegially discussed and explained to patients in advance. A robotic (Da vinci, Intuitive, Sunnyvale, CA, USA) or laparoscopic approach was chosen by surgeons according to BMI and the type of resection/excision planned. We also evaluated the total resection of endometriosis (excision of all lesions) compared to conservative surgery in this study. Conservative surgery was the targeted resection of endometriosis with a left residual endometriosis to minimise post-operative complications in case of frozen pelvis, rectal risk, or urinary risk. The choice between total and conservative surgery was collegially discussed by experienced surgeons. We chose conservative surgery (ConservS) for patients in whom there was a risk of severe complications if complete surgery was performed (colorectal or urinary tract involvement). There was no evaluation of post-operative remaining tissue by MRI or US assessment. The type of surgery was explained in advance to patients.

The data were collected from medical records, namely patient characteristics (age, BMI, history of prior surgery for endometriosis or other, pre- and post-operative painkiller and hormonal treatment), pre- and post-operative pain evaluation (Visual Analogue Scale (VAS) for dysmenorrhea and dyspareunia, use of level 3 painkillers), type of surgery (RL or CL), conservative (ConservS) or total surgery (TS), and type of resection (adhesiolysis, hysterectomy, salpingectomy, ureterolysis, utero-sacral ligament (USL) resections, rectal shaving, vesical node resection). Total hysterectomies were performed for patients around the age of menopause who did not want additional children and who asked for this type of intervention. Surgical and post-operative complications were reported using the Clavien–Dindo classification [15]. The rate of pain recurrence (reoccurrence of painful symptoms with or without imagery evidence: MRI and US assessment) was calculated as a percentage. Quality of life was evaluated using EHP-5 questionnaires (pre- and 2-years post-operative evaluation).

### 2.2. Quality of Life Evaluation

The EHP-5 (Endometriosis Health Profile) is a short version of the EHP-30 (HRQoL: Health-Related Quality of Life), a specific questionnaire about quality of life in cases of endometriosis. It was recently validated in French [14]. The EHP-5 addresses the patient’s social and sexual life, daily life management of pain, and impact on work, emotions, relationship, and potential infertility with 11 questions. The answer to each questions involves five levels ranging in order of severity, as “never”: 0, “rarely”: 25, “sometimes”: 50, “often”: 75, and “always”: 100. The higher the score, the greater the effect on quality of life. The best score possible for each question was 0, and the worst score was 100. The EHP-5 is shown in Figure 1.

### 2.3. Statistics

Continuous data are presented as median (min–max) and compared with the Mann–Whitney test. Categorical variables are presented as a number (percentage) and were compared with a Chi-square test or Fisher’s exact test, as appropriate.

The evolution of EHP-5 scores was estimated pre- and post-operation using Wilcoxon’s signed rank paired test; the difference for the EHP-5 score between groups of patients was calculated as (post EHP–pre EHP)/pre EHP, and the comparison between groups was performed with a Mann–Whitney test. Statistical analyses were performed using SAS software (version 9.4; SAS Institute, Cary, NC, USA). A *p* value < 0.05 was considered statistically significant.

### 2.4. Ethics

Informed consent was obtained from all participants in the study. The study design was validated by the Ethics Committee of Foch Hospital on 8 December 2021 (registration number: IRB00012437).

## 3. Results

We identified 95 patients who met the inclusion criteria. We had complete data for pre- and 2-years post-operative EHP-5 questionnaires and VAS (Visual Analogic Scale) for dysmenorrhea and dyspareunia for 54 of the 95 patients. A flowchart is shown in Figure 2.

### 3.1. Global Population Results

We had a 100% follow-up rate by including retrospectively only patients with complete data for the main objective. With a two-sided significance level of alpha = 0.05, the power of the study was 88.36% for the main objective. An important decrease in 2-year post-operative EHP-5 scores was found in the global population (pre-op: 61.36 (42.18–68.75) and 2-year post-op: 20.45 (0–38.06); *p* < 0.001). The VAS for dysmenorrhea was also lower (pre-op: 8 (7–9.75) vs. 2-year post-op: 3 (2–5.25); *p* < 0.001), as was the VAS for dyspareunia (pre-op: 6 (3.1–8.9) vs. 2-year post-op: 3 (0–6); *p* < 0.001) (Figure 3).

Included patients had intense pain symptoms with a pre-operative VAS for dysmenorrhea and dyspareunia of 8.18 (7–9.75) and 6 (4–8) out of 10, respectively. The majority of patients took hormonal treatment, and 46 (85%) were treated at least three months before surgery. Nine patients (16.7%) had rectal shaving (including six rectal resections initially planned in other centres). One laparoconversion was performed (haemorrhage on a voluminous peri-ureteral endometriosis node), and two Clavien 3b complications were reported (one eventration and one pelvic abscess). Less than half the patients (17, 31.4%) had a post-operative hormonal treatment. A third of the population, 16 out of 52 patients (30.7%), still depended on level 3 painkillers 2 years after surgery. Of the 54 patients, 6 (11.1%) had a recurrence of pelvic pain. The results are presented in Table 1.

### 3.2. Subgroup Analysis

Two subgroups were studied to explore different aspects of surgical management: CL compared to RL, and TS compared to ConservS.

The first subgroup studied, CL vs. RL, was comparable for type of resection and BMI, but patients were older in the RL group (RL: 36.5 years (29.75–43.5) vs. CL: 34 years (27.5–37.5); *p* = 0.04). The post-operative EHP-5 score was significatively improved in the RL group (pre-op: 65.9 (59.09–71.02) vs. 2-year post-op: 11.4 (0–38.06)) and in the CL group ((pre-op: 50/100 (34.65–68.18); 2-year post-op: CL 27.27 (14.20–40.34)). There was a difference in favour of RL when comparing 2-year post-operative EHP-5 scores between RL and CL (*p* = 0.04). The VAS for post-operative dysmenorrhea was lower in the RL compared to the CL group (RL: 3(0–5)/10 vs. CL: 5(2–7)/10); *p* = 0.04). Hysterectomy and utero sacral ligament (USL) resections rates were also higher in the RL group (hysterectomy—RL: 10/26 (38.5%) vs. CL: 0/28 and *p* < 0.001; USL resection—RL: 17/26 (31.8%) vs. CL: 11/28 (9.4%) and *p* = 0.03). A difference was found in the duration of analgesia, which was higher in RL than CL (RL: 155 (119–184.5 min) vs. CL 102 (90.75–134.5) minutes; *p* = 0.05). No difference was found for the recurrence rate of painful symptoms comparing CL and RL groups (CL: 17.8% (5/28) and RL: 3.8% (1/26); *p* = 0.19). The results are presented in Table 2 and Figure 3.

An improvement in the EHP-5 score was also found in the ConservS (pre-op: 61.4 (38.06–71.59), 2-year post-op: 22.7 (11.93–38.07)) and TS groups (pre-op: 61.59 (51.70–68.75), 2-year post-op: 13.63 (0–44.30)), with no difference between the two groups for 2-year post-operative EHP-5 scores (*p* = 0.23). Two-year post-operative VAS for dysmenorrhea was similar in ConservS vs. TS (ConservS: 3 (0.25–2)/10, TS 3.5 (0–4.5)/10; *p* = 0.9). There was more adhesiolysis in the ConservS group (ConservS: 71.9% (23/32) vs. TS: 40.91% (9/22); *p* = 0.02). USL resection rates were higher in the TS group (TS: 95.45% (21/22) vs. ConservS: 21.9% (7/32); *p* < 0.001). Likewise, hysterectomy resections were also higher in the TS group (TS: 31.82% (7/21) vs. ConservS: 9.4% (3/32); *p* = 0.04). No difference was found for the recurrence rate of pain between the ConservS and TS groups (ConservS: 15.6% (5/32) and TS: 4.5% (1/22); *p* = 0.38). All results are presented in Table 1 and Table 2 and Figure 4.

## 4. Discussion

Endometriosis patients who underwent surgery in our centre had an altered quality of life and an elevated VAS for dysmenorrhea and dyspareunia, indicating very painful symptoms after the failure of all medical treatment.

Our results showed an improvement in quality of life based on the EHP-5 questionnaire, 2 years after minimally invasive surgery for painful DIE. As shown by Aubry and Al [16], the EHP-5 questionnaire is an effective and sensitive tool with a suitable response rate and is accurate regarding symptoms of endometriosis. The minimally invasive approach enabled an important decrease in VAS for dysmenorrhea and dyspareunia 2 years after surgery, proving the efficiency of surgery after the failure of medical treatment [8,9,17]. In fact, we know that impaired quality of life is linked to chronic pelvic pain, and it affects social, professional, psychological, and sexual life [18]. A significant decrease in these symptoms is another point in favour of surgical management [8]. Mini-invasive surgery is a suitable option to relieve pain when medical and hormonal treatments are no longer efficient [19,20]. It also appeared that conservative surgery was as efficient as total surgery regarding QoL on a mid-term basis.

Although the study lacked power, with no preliminary effective calculation and non-comparable groups, the subgroups analysis showed that there was a greater improvement in quality of life, slightly greater in RL compared to the CL group. Most studies show no difference between the two courses of surgery except for operative length, which tends to be longer for robotic laparoscopy, and a tendency of decreased bleeding with robotic surgery [21]. The cost of robotic surgery also needs to be balanced with the surgeon’s comfort [22]. Nevertheless, robotic laparoscopy has been used more and more frequently since its commercialisation in 2000 [23], and it seems to be suitable for complex surgery, such as deep endometriosis surgical treatment, with its “wrist like” motions, better precision, mobility, dexterity, and 3D vision, which increases the surgeon’s view of the operative field [24]. It also has a shorter learning curve compared to CL surgery [12]. This is the first study showing that robotic laparoscopy has a tendency for superiority, but these results must be balanced with the limitations of our study, which was retrospective. Indeed, the groups were not comparable, especially concerning the rate of hysterectomy, which was higher in RL and could be a major bias.

The total excision of endometriosis is recommended by all international health organisations [1,4,5]. Surprisingly, conservative treatment showed adequate results for the evolution of quality of life in our study for patients 2 years after surgery, which goes against current knowledge [1,19,20]. We performed rectal shaving in 20% of patients to avoid rectal resection, which is known to cause complications (rectovaginal fistula, anastomotic stenosis, voiding dysfunction) [5,8]. We also avoided bilateral USL resection, which can cause long-term persistent urinary retention (UR) requiring self-catheterisation [20], but we did not specifically collect post-operative rates of incontinence and prolapse. As a result, we had a low rate of complications [25]. The problem with conservative surgery is the risk of recurrence of symptoms linked to the lesions and cells left in place, estimated at 15% for conservative surgery vs. 3% for total surgery after 5 years [26]. Even if we did not observe a significant difference in the rate of painful symptom recurrence comparing conservative and complete surgery, the rate was clinically meaningful in the CL group and should be explained to patients before surgery. Furthermore, a third of patients still needed level 3 painkillers and hormonal treatment after surgery, showing that management of endometriosis required a holistic approach in order to be effective alongside surgery (painkillers, hormonal treatments, diet, and lifestyle rules) [1]. Moreover, a significant percent of patients chose to stop hormonal treatment after surgery, with explanations including significant pain decrease, pregnancy desire, and side effects of hormonal treatment. Other studies will be necessary to confirm these findings, but the interest of a partial surgery in DIE for the improvement of the quality of life in association with other treatments seems to be an important result of our work.

The strength of this study lies in assessing quality of life using a specific questionnaire validated for endometriosis at the mid-term after surgery. The limitations were that we used retrospective data, which have known associated biases such as loss of follow-up and non-comparable groups. We also had a small effective calculation, with no preliminary power calculation, which strongly limited our conclusions. Patients who had taken a laparotomy route were also excluded. Follow-up was limited to 2 years, with no long-term data. Moreover, complications of prolapsus and persistent urinary retention were not collected. Prospective evaluation with a larger sample size is necessary for future studies. Of course, the questions of preserved fertility and DIE remaining in place after surgery are important and were not explored in this study.

Questions about robotic vs. classic laparoscopy still need to be answered to confirm this trend with regard to the balance between surgeon comfort and medico-economic cost of robotic surgery.

## 5. Prospects

Surgical treatment of endometriosis needs a specific strategy for each patient. Today, it is clear that every surgical treatment has issues (on a short-, mid-, and long-term basis). More data are necessary to compare the benefits of the robotic vs. laparoscopic method, as well as conservative and total surgery for deep painful endometriosis. A prospective evaluation with a greater effective calculation and a long-term follow-up that evaluates fertility, the recurrence rate of pain, and the quality of life is mandatory.

## 6. Conclusions

Minimally invasive surgery for patients with pain and deep infiltrating endometriosis after the failure of medical treatment led to an improved quality of life and a significant decrease in dysmenorrhea and dyspareunia 2 years after surgery. Conservative surgery showed comparable results to total surgery. The optimal route for surgery as regards to robotic and classic laparoscopy remains unclear and needs to be assessed with a larger sample and in a prospective way.

## Figures and Tables

**Figure 1 jcm-11-06132-f001:**
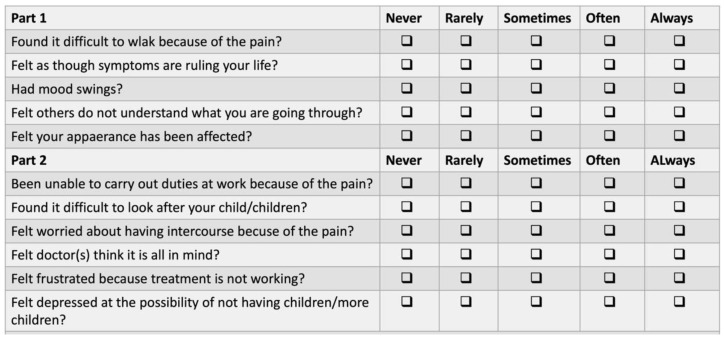
EHP-5 questionnaire [15].

**Figure 2 jcm-11-06132-f002:**
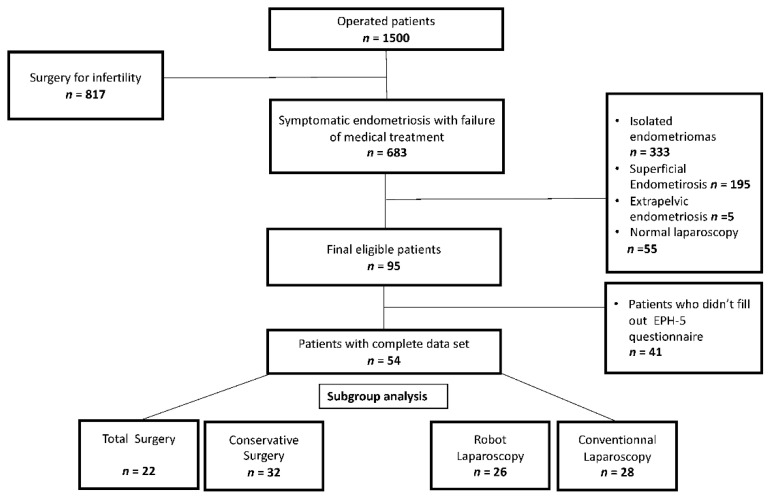
Flowchart of the study.

**Figure 3 jcm-11-06132-f003:**
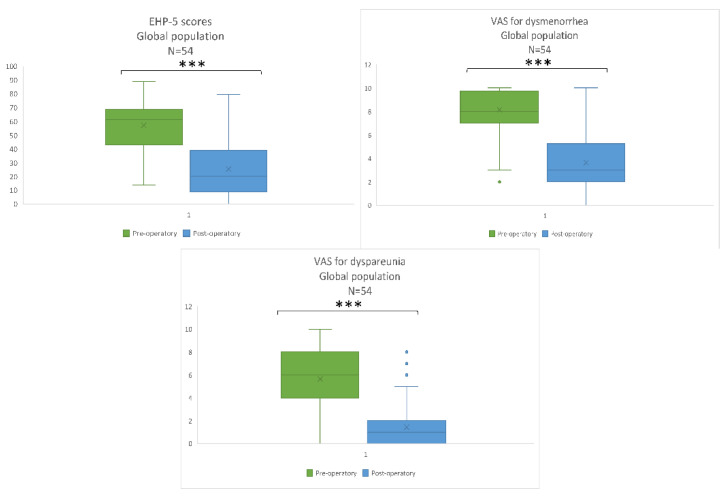
Boxplot for pre- and 2-year post-operative patients for EHP-5 score and VAS for dysmenorrhea and dyspareunia in the global population. Each boxplot shows the median and 1st and 3rd quartiles, and minimal and maximal values. ° represents missing data. *** *p* < 0.001.

**Figure 4 jcm-11-06132-f004:**
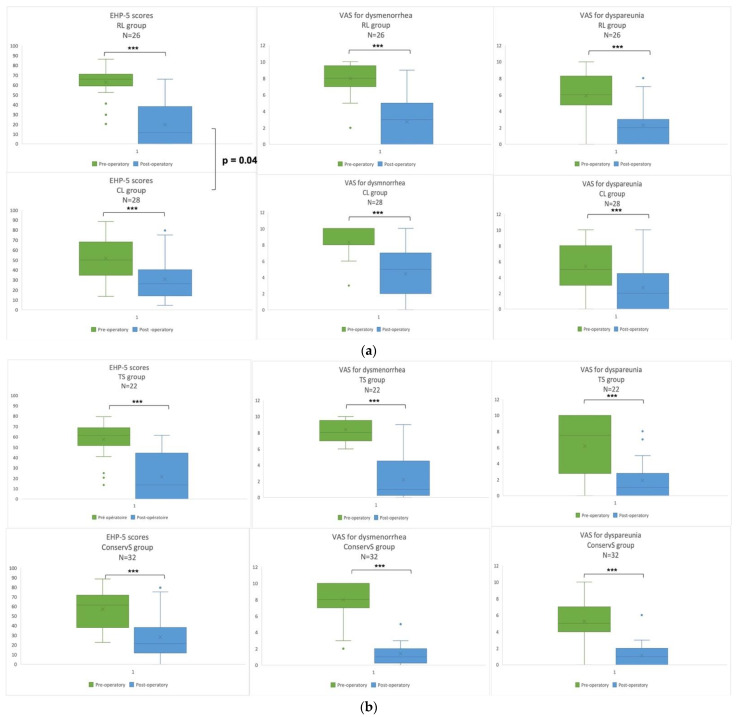
(**a**) Boxplot for pre- and 2-year post-operative RL and CL groups. Each boxplot shows the median and 1st and 3rd quartiles, and minimal and maximal values. ° represents missing data. *** *p* < 0.001. (**b**) Boxplot for pre- and 2-years post-operative EHP scores, VAS for dysmenorrhea, and VAS for dyspareunia for ConservS and TS groups. Each boxplot shows the median and 1st and 3rd quartiles, and minimal and maximal values. ° represents missing data. *** *p* < 0.001.

**Table 1 jcm-11-06132-t001:** Description of global population using median (min–max) and % for age (years), BMI (kg/m^2^), pre-operative treatment (morphine derivative painkiller, hormonal treatment), prior surgery for endometriosis, and type of resection in the two subgroups: RL vs. CL and ConservS vs. TS.

Studied Variables	Global Population *n* = 54	CL *n* = 28	RL *n* = 26	*p*	ConservS *n* = 32	TS *n* = 22	*p*
Age (years) Median (min–max)	35.5 (27.8–43.2)	34 (27.5–37.5)	36.5 (29.75–43.5)	**0.04**	37 (29–39)	34 (27.75–39)	0.83
BMI (kg/m^2^) Median (min–max)	23 (18.7–27.3)	23 (21–29)	23 (20.5–27.5)	0.27	24.5 (21–28.25)	21.5 (20–26.5)	0.21
Pre-operative hormonal treatment (%)	80 (43/54)	78 (22/28)	81 (21/26)	1	72 (23/32)	91% (20/22)	0.16
Level 3 painkiller (%)	20.4 (11/54)	17.9 (5/28)	26.9 (7/26)	0.72	25 (8/32)	18.18 (4/22)	0.20
Prior surgery for endometriosis (%)	11.1 (6/54)	14.3 (4/28)	7.7 (2/26)	0.25	12.5 (4/32)	9.1 (2/22)	0.31
Hysterectomy (%)	18.5 (10/54)	0 (0/28)	38.5 (10/26)	**<0.001**	9,4 (3/32)	31.82 (7/21)	**0.04**
Salpingectomy (%)	11.1 (6/54)	10.7 (3/28)	11.5 (3/26)	0.99	15.6 (5/32)	4.55 (1/22)	0.38
Ureterolysis (%)	31.5 (17/54)	28.6 (8/28)	34.6 (9/26)	0.83	25 (8/32)	40.91 (9/22)	0.36
Adhesiolysis (%)	50 (27/54)	64.3 (18/28)	53.8 (9/26)	0.43	71.9 (23/32)	40.91 (9/22)	**0.02**
USL resection (%)	51.9 (28/54)	39.3 (11/28)	65.4 (17/26)	**0.03**	21.9 (7/32)	95.45 (21/22)	**<0.001**
Bladder node resection (%)	16.7 (9/54)	7.1 (5/28)	7.7 (4/26)	0.94	6.2 (2/32)	9.09 (2/22)	0.69
Superficial digestive node resection (%)	18.5 (10/54)	25 (7/28)	26.9 (3/26)	0.87	31.2 (10/32)	18.18 (0/22)	0.28

**Table 2 jcm-11-06132-t002:** Secondary outcomes and post-operative description for global population, robotic laparoscopy (RL), classical laparoscopy (CL), total surgery (TS), and conservative surgery (ConservS).

Studied Variables	Global Population *n* = 54	CL *n* = 28	RL *n* = 26		ConservS *n* = 32	TS *n* = 22	*p*
Post-operative hormonal treatment (%)	31.5	25 (7/28)	38 (10/26)	0.38	34.3 (11/32)	27.2 (6/22)	0.77
Post-operative use of level 3 pain killer (%)	30.7 (16/52)	50 (7/28)	32.14 (9/28)	0.55	43.75 (14/32)	22.72 (5/22)	0.15
Pre-operative complications (%)	2 (1/53)	3.7 (1/27)	0 (0/26)	0.32	3.45 (1/28)	0 (0/22)	0.38
Post-operative complications (%)	5.7 (3/52)	7.69 (2/26)	3.85 (1/26)	0.55	6.45 (2/31)	4.76 (1/21)	0.8
Blood loss 400 (mL)	0.02 (1/52)	0	4.17 (1/24)	0.33	3.85 (1/26)	0 (0/22)	0.41
Post-operative recurrence rate (%)	11.1% (6/54)	17.8 (5/28)	3.8 (1/26)	0.19	15.6 (5/32)	4.5 (1/22)	0.38
Complete surgery (%)	40.7	17.8 (7/28)	65.38 (15/26)	**0.04**	**NA**	**NA**	**NA**

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
