# Peer review of "Improvement of Post-Operative Quality of Life in Patients 2 Years after Minimally Invasive Surgery for Pain and Deep Infiltrating Endometriosis"

_jcm, 2022, doi:10.3390/jcm11206132_

Round 1

Reviewer 1 Report

With this retrospective analysis, the authors examined the QoL of the patients -the article with symptomatic DIE after a two-year follow-up period of endometriosis MI-surgery. Using the EHP-score, a sustained improvement in the symptoms could be determined in the study. The research question of the study is clinically highly relevant. However, the study needs some major and minor changes before it is published. The following major and minor changes and additions are necessary:   1-) The authors divided the cohort into 4 groups, although herefor two groups (CL and RL) examine surgical route and another two groups (ConsevS and TS) examine surgical strategy for endometriosis treatment. The distribution of the patients into 4 groups, and the initial advertisement, easily lead to misunderstandings when reading the manuscript. Only when reading the results does it become clear that this is about two subgroup analysis. Therefore, the manuscript should be formulated correctly at the beginning so that thoughts about incorrect methodology do not quickly arise. In addition, the aim of the analysis should be defined again and precisely, whether the role of the type of MIS (CL vs. RL) or the role of the surgical strategy (complete vs. incomplete resection) of DIE is to be examined.   2-) In Results, line 175, a significantly lower VAS for dysmenorrhea was found in RL group compared to CL group. The 18.5% of the total patients underwent a hysterectomy. 100% of the hysterectomized patients are in the group of RL. Furthermore, median age is significantly high in the RL group. This can make a statistical difference, but clinically it makes little sense to compare the two groups in this regard. One reason for this could be that the older patients were indicated for a hysterectomy via RL. A selection bias already arises for this. The difference also influences other factors such as duration of surgery, etc. and postoperative outcome. Therefore, it is difficult to accept the RL as being advantageous over the CL as presented in Discussion (lines 227-228). A further table with surgical procedures for all three groups (all, RL group, CL group) would be necessary for this in order to create an overview of the extent of the surgical procedures in the groups. In this manuscript there is a good chance to focus on the radicality of DIE surgery and compare both groups (ConservS vs. TL). The finding that there is no significant difference between the ConservS and TL groups is a major finding. In this subgroup analysis, there would then be little selection bias, as can also be seen in Table 2.   minor changes   3-) The abbreviation for Endometriosis Health Profile is described in the literature as EHP. For consistency in the literature, the abbreviation EPH should be rewritten as EHP in manuscript where it appears. The abbreviations should also be explained in its first appearance, such as in the line 247 RPM.   4-) The authors have linked some references from the guidelines or review. Correct references and authors should be cited directly for this (e.g. line 255).   5-) The conclusion part is missing in the abstract.   6-) Moreover, some important clinical data of the follow-up period are missing in manuscript. For example, median follow-up, parity, type of hormonal treatment, etc. The factors contributing to 2-year postoperative QoL evaluated how many patients had incontinence and prolapse symptoms after resection of LUS assumed that 51.8% of patients had LUS resected.   7-) The authors state that only 31.4% of the patients received postoperative hormonal therapy. These low percentages can negatively impact the results. For this purpose, possible hypotheses should be explained in the manuscript, for which reasons the patients did not want any hormonal therapy postoperatively, although 85% of the patients had hormonal therapy before the surgery.   8-) Bladder and superficial digestive node resections were described in the cohort, 16.7% and 18.5% respectively. In addition, 60% of the patients were in the ConservS group. Is it true that no bladder or bowel resections with anastomoses were performed for DIE between 2010 and 2019?   Are the menifestations of the endometriotic lesions of the patients described with any classification? If so, what classification was used for it?   9-) In line 181, actually Table 2, presents the results of the 2nd subgroup analysis (Concerv vs. TL), and not the first (RL vs. CL). Please provide the appropriate link to the text here.   10-) Finally, improving the language of the manuscript and avoiding poetic words (e.g. use important decrease instead of significant decrease in lines 26, 129 etc) is recommended.

Author Response

Reviewer 1

Answers to reviewer 1:

We would like to thank reviewer 1 for his accurate look on this work and his remarks which will help to provide a better article.

Point 1).

 The authors divided the cohort into 4 groups, although therefore two groups (CL and RL) examine surgical route and another two groups (ConservS and TS) examine surgical strategy for endometriosis treatment. The distribution of the patients into 4 groups, and the initial advertisement, easily leads to misunderstandings when reading the manuscript. Only when reading the results, it becomes clear that this is about two subgroup analysis. Therefore, the manuscript should be formulated correctly at the beginning so that thoughts about incorrect methodology does not quickly arise. In addition, the aim of the analysis should be defined again and accurately, whether the role of the type of MIS (CL vs. RL) or the role of the surgical strategy (complete vs. incomplete resection) of DIE is to be examined.

Response 1).

 Thank you for your comment, in fact we studied two groups, and it’s not accurate to talk about four sub-groups.

The two subgroups studied were:

- Robotic Laparoscopy vs Classic Laparoscopy

- Conservative vs Total Surgery

We changed it in the abstract and throughout the whole article: for example, lines 24-26: "EHP-5 scores were complete for 54 patients and two subgroups were analyzed: Classic laparoscopy (CL) vs Robot Laparoscopy (RL) and Conservative Surgery (ConservS) vs Total Surgery (TS). and lines 163-164: " Two sub-groups were studied to explore different aspects of surgical management: CL compared to RL, and TS compared to ConservS".

We also changed the flow-chart in this way, to understand that we studied the same population in two ways: on one hand, the type of surgery (Robotic vs classic laparoscopy) and on the other hand, type of resection (Conservative Surgery and Total Surgery).

Figure 2. Flow-chart of the study

About the main objective of the study, we chose to compare the two different types of minimally invasive surgery: robotic laparoscopy vs classic laparoscopy. But we extended the main objective in the introduction following your recommendation by adding the two types of surgery specificly.

We mentioned it line 59-61: "The principal aim of this study was the improvement of quality of life using EHP-5 [15] 2 years after minimally invasive surgery for painful DIE: subgroups comparing robotic vs classic laparoscopy and total versus conservative surgery were also analyzed".

Point 2).

In Results, line 175, a significant decrease VAS for dysmenorrhea was found in RL group compared to CL group. The 18.5% of the total patients underwent a hysterectomy. 100% of the hysterectomized patients were in the group of RL. Furthermore, median age wassignificantly high in the RL group. This could make a statistical difference, but clinically it maked little sense to compare the two groups in this regard. One reason for this could be that the older patients were indicated for a hysterectomy via RL. A selection bias already arose for this. The difference also influenced other factors such as duration of surgery, etc. and postoperative outcome. Therefore, it was difficult to accept the RL as being advantageous over the CL as presented in Discussion (lines 227-228). A further table with surgical procedures for all three groups (all, RL group, CL group) would be necessary to this in order to create an overview of the extent of the surgical procedures in the groups.

In this manuscript, there was a good chance to focus on the radicality of DIE surgery and compared both groups (ConservS vs. TL). The finding was that there was no significant difference between the ConservS and TL groups was  a major finding. In this subgroup analysis, there would then be little selection bias, as could  also be seen in Table 2.

Response 2).

Once more, thanks for this comment, you are right, we also thought that a higher rate of hysterectomy in Robotic group could be a bias compared to classic laparoscopy.

Concerning, robotic vs classic laparoscopy, we choose to speak about a "slight" advantage of robotic surgery: line 29-32 " For sub-groups analysis: EHP-5 scores were improved in RL groups (pre-op: 65.9 (59.09–71.02) vs 2-y post-op: 11.4 (0–38.06); p<.001) and CL group (pre-op: 50 (34.65-68.18) vs 2-y post-op: 27.27 (14.20-40.90); p<.001) with a slight advantage for RL (p=.04)".

But we reconsidered these assertions by insisting on the different bias: non-comparability of groups and the necessity of confirmation of these results with more robust studies in the discussion: line 230-234: " This is the first study showing that robot laparoscopy has a tendency to a superiority, but these results must be balanced with the limitations of our study which was retrospective. Indeed, the groups were not comparable especially concerning the rate of hysterectomy, which was higher in RL and could be a major bias". 

Concerning the table, you are right, we complete table 1 with details about each subgroup.

Studied variables

Global

Population

N=54

CL

N= 28

RL

N= 26

p

ConvervS

N=32

TS

N=22

p

Age (Years)

Median (min-max)

35.5 (27.8-43.2)

34 (27.5-37.5)

36.5 (29.75- 43.5)

.04

37 (29-39)

34 (27.75-39)

.83

BMI (kg/m2)

Median (min-max)

23 (18.7-27.3)

23 (21-29)

23 (20.5-27.5)

.27

24.5 (21-28.25)

21.5 (20-26.5)

.21

Pre-operative hormonal treatment (%)

80 (43/54)

78 (22/28)

81 (21/26)

1

72 (23/32)

91% (20/22)

.16

Painkiller levell III (%)

20.4 (11/54)

17.9 (5/28)

26.9 (7/26)

.72

25 (8/32)

18.18 (4/22)

.20

Prior surgery for endometriosis (%)

 11.1 (6/54)

14.3 (4/28)

7.7 (2/26)

.25

12.5 (4/32)

9.1 (2/22)

.31

Hysterectomy (%)

18.5 (10/54)

0 (0/28)

38.5 (10/26)

<.001

9,4 (3/32)

31.82 (7/21)

.04

Salpingectomy (%)

11.1 (6/54)

10.7 (3/28)

11.5 (3/26)

.99

15.6 (5/32)

4.55 (1/22)

.38

Ureterolysis (%)

31.5 (17/54)

28.6 (8/28)

34.6 (9/26)

.83

25 (8/32)

40.91 (9/22)

.36

Adhesiolysis (%)

50 (27/54)

64.3 (18/28)

53.8 (9/26)

.43

71.9 (23/32)

40.91 (9/22)

.02

USL  resection (%)

51.9 (28/54)

39.3 (11/28)

65.4 (17/26)

.03

21.9 (7/32)

95.45 (21/22)

<.001

Bladder node resection (%)

16.7 (9/54)

7.1 (5/28)

7.7 (4/26)

.94

6.2 (2/32)

9.09 (2/22)

.69

Superficial digestive node resection (%)

18.5 (10/54)

25 (7/28)

26.9 (3/26)

.87

31.2 (10/32)

18.18 (0/22)

.28

Table 1. Description of global population using median (min-max) and % for age (years), BMI (kg/m2), pre-operative treatment (morphine derivative painkiller, hormonal treatment), prior surgery for endometriosis, type of resection in the two subgroups: RL vs CL and ConservS vs TS.

Thanks for your support, it is clearly a big finding that quality of life is improved after Conservative surgery with a lasting mid-term effect (two years), compared to total surgery.

We rephrase a sentence to emphase this point: line 251-253: " Other studies will be necessary to confirm these findings, but the interest of a partial surgery in DIE for the improvement of the quality of life in association with other treatments seems to be an important result of our work.”

Point 3).

The abbreviation for Endometriosis Health Profile is described in the literature as EHP. For consistency in the literature, the abbreviation EPH should be rewritten as EHP in manuscript where it appears. The abbreviations should also be explained in its first appearance, such as in the line 247 RPM.  

Response 3).

Once more again, thank you for your accurate look on this article, EPH-5 is the French version of EHP-5, it's clearly an omission. As asked, we changed it, to describe it at its first appearance line 21 and throughout the article.

Same for RPM who is a French translation for persistent urine residue, line 260 has been changed to persistent urinary retention which is the accurate term in English.

Point 4).

The authors have linked some references from the guidelines or review. Correct references and authors should be quoted directly for this (e.g., line 255).  

Response 4).

Thank you again for your precision, we changed it to precisely quote the paragraph and line of our references:

-Line 304 ESHRE recommendations 2022: The members of the Endometriosis Guideline Core Group, Becker CM, Bokor A, et al. ESHRE guideline: endometriosis. Human Reproduction Open 2022;2022:hoac009.

- Line 330 for ESHRE recommendations 2014:Dunselman GAJ, Vermeulen N, Becker C, et al. ESHRE guideline: management of women with endometriosis. Human Reproduction 2014;29:400‑12.

Point 5).

The conclusion part is missing in the abstract.

Response 5).

You are right, we noticed ours results without conclusion.

We added a sentence to end it, line 33-35: " Minimally invasive surgery achieved to improve QoL for DIE patient 2-year after surgery and conservative surgery showed comparable results to total surgery".

Point 6).

In addition, some important clinical data of the follow-up period are missing in the manuscript. For example, median follow-up, parity, type of hormonal treatment, etc. The factors contributing to 2-year postoperative QoL evaluated how many patients had incontinence and prolapse symptoms after resection of LUS assumed that 51.8% of patients had LUS resected.

Response 6).

All these remarks are relevant, however, because of missing data for post-operative parity, incontinence and prolapsus symptoms after USL resections, we couldn’t analyze them.

We modified our discussion to talk about these limitations: lines 230-234" This is the first study showing that robot laparoscopy has a tendency to a superiority, but these results must be balanced with the limitations of our study which was retrospective. Indeed, the groups were not comparable especially concerning the rate of hysterectomy, which was higher in RL and could be a major bias”. 

And lines 255-260: " The limitations were that we used retrospective data, which involves known associated bias such as loss of follow-up and non-comparable groups. We also had a small effective, with no preliminary power calculation, which strongly limited our conclusions. Patients who had taken a laparotomy route were also excluded. Follow-up was limited to 2 years, with no long-term data. Also, complications about prolapsus and persistent urinary retention were not collected”.

2 years follow-up was 100% in our population because we decided to include only patients who had complete clinical data and EHP-5 scores 2 years after surgery.

We added a line to explain the good follow-up on the main objective, lines 140-141: "We had a 100% follow-up by including retrospectively only patients with complete date for the main objective".

Point 7).

The authors says that only 31.4% of the patients received postoperative hormonal therapy. These low percentages can negatively impact the results. For this purpose, possible hypothesis should be explained in the manuscript, for which reasons the patients did not want any hormonal therapy postoperatively, although 85% of the patients had hormonal therapy before the surgery.  

Response 7).

For post-operative hormonal treatment, it appears that patient didn't want hormonal treatment because of the decrease of pain (which was manageable with pain killer or nothing), intention to be pregnant, and for some decided to stop pills because of side effects of hormonal treatment (mood swings, weight gain, decreased libido...). We explained that lines 249-251:

" Also, a significant percent of patients has chosen to stop hormonal treatment after surgery, several explications can be provided:  significant pain decrease, pregnancy desire and for some due to side effects of hormonal treatment...."

Point 8).

Bladder and superficial digestive node resections were described in the cohort, 16.7% and 18.5% respectively. In addition, 60% of the patients were in the ConservS group. Is it true that no bladder or bowel resections with anastomoses were performed for DIE between 2010 and 2019?  

Are the manifestations of the endometriotic lesions of the patients described with any classification? If so, what classification was used for it?

Response 8).

We would like to thank you once more for your pertinent approach of our subject, which wasn't easy to aboard.

During this period, only one bladder resection occurred, it was a complex surgery, and laparotomy route was preferred. Four digestive resections with anastomosis were performed but laparotomy was also preferred.

Lines 73-75: " Patients who followed a laparotomy route (5 patients: 1 complex vesical resection and 4 digestive resections with anastomosis) and patients who didn't fill EHP-5 questionnaires were also excluded.”

For classification of endometriosis lesions, we choose to use the revised American fertility society classification of endometriosis (AFSr) . In this study all patients had AFSr scores >40.  

Lines 76-77: " All patients had AFSR (revived American Fertility Society ) scores > 40 points (stage 4). ".

Point 9).

In line 181, actually Table 2, presents the results of the 2nd subgroup analysis (ConservS vs. TL), and not the first (RL vs. CL). Please provide the appropriate link to the text here.

Response 9).

Thank you for the level of accuracy of your comments, in fact, we decided to add new tables with data for ConservS vs Total Surgery and Robotic vs Classic laparoscopy surgery.

Results for each group are presented table 2.

Studied variables

Global

Population

N=54

CL

N= 28

RL

N= 26

ConservS

N=32

TS

N=22

p

Post-operative hormonal treatment

(%)

31.5

25 (7/28)

38 (10/26)

.38

34.3 (11/32)

27.2 (6/22)

.77

Post-operative use of level 3 pain killer (%)

30.7 (16/52)

50 (7/28)

32.14 (9/28)

.55

43.75 (14/32)

22.72 (5/22)

.15

Per-operative complications (%)

2 (1/53)

3.7 (1/27)

0 (0/26)

.32

3.45 (1/28)

0 (0/22)

.38

Post-operative complications (%)

5.7 (3/52)

7.69 (2/26)

3.85 (1/26)

.55

6.45 (2/31)

4.76 (1/21)

.8

Blood loss

1.   400 (mL)

.02 (1/52)

0

4.17 (1/24)

.33

3.85 (1/26)

0 (0/22)

.41

Post-operative reccurence rate

(%):

11.1% (6/54)

17.8 (5/28)

3.8 (1/26)

.19

15.6 (5/32)

4.5 (1/22)

.38

Complete surgery (%)

 40.7

17.8 (7/28)

65.38 (15/26)

.04

NA

NA

NA

Table 2. Secondary outcomes and post-operative description for global population and Robotic surgery (RL) vs Classic Laparoscopy (CL) and TS (total surgery) vs conservative surgery (ConservS), post-operative treatments (morphine derivative painkiller, hormonal treatment), and recurrence rate of painful symptoms.

Point10).

Finally, improving the language of the manuscript and avoiding poetic words (e.g. use important decrease instead of significant decrease in lines 26, 129 etc) is recommended.

Response 10).

Thank you again, for helping us improving the quality of our article. For English revisions, we ask an English native speaker to improve the English.

For example

- line 26-27: " In global population, an important decrease of 2-year post-operative EHP-5 (Endometriosis Health Profile-5) scores was found"

- line 147-148: " A majority of patient took hormonal treatment and 46 (85%) were treated at least three months before surgery".

Reviewer 2 Report

1. The need to present immediate postoperative complications

2. Evaluation of surgical excision and remaining endometriotic tissue

3. Possible histo-pathological surprises on the resection samples?

4. The quality of life assessed in the standard, non-endometriosis population, what values ​​does it have?

5. Have other concurrent pathologies been identified at the gynecological level?

6. Some comparative discussions of the therapeutic options appear necessary

Author Response

Reviewer 2

Answers to reviewer 2:

We would like to thank reviewer 2 for his precious comments on our article and the time allowed to help us to improve the final version.

Point 1).

The need to present immediate postoperative complications

Response 1).

Post-operative complications were a rare event and are detailed line 150-152: " One laparoconversion occurred (hemorrhage on a voluminous peri-ureteral endometriosis node), and two Clavien 3b complications were reported (one eventration and one pelvic abscess)".

Point 2).

Evaluation of surgical excision and remaining endometriotic tissue.

Response 2).

For conservative surgery, we explained our methodology in the material and in method section, we add a sentence for absence of evaluation of remaining endometriosis tissue.

Lines 82-90: " We also evaluated the total resection of endometriosis (excision of all lesions) compared to conservative Surgery in this study. Conservative surgery was the targeted resection of endometriosis with a left residual endometriosis to minimise post-operative complications in case of frozen pelvis, rectal or urinary risk. The choice between total and conservative surgery was collegially discussed by experienced surgeons. We chose conservative surgery (ConservS) for patients in whom there was a risk of severe complications if complete surgery was performed (colorectal or urinary tract involvement). There was no evaluation of post-operative remaining tissue by MRI or US assessment.”

Point 3).

Possible histopathological surprises on the resection samples?

Response 3).

All the anatomopathological analysis were collected. No other disease than endometriosis tissue was found. We added a sentence about it line 79-80: “No case of misdiagnosis or especially malignant tumor was revealed on histopathology analysis.

Point 4).

 The quality of life assessed in the standard, non-endometriosis population, what values ​​does it have?

Response 4).

EHP -5 were assessed on endometriosis patients to compare the evolution of QoL before and after surgery.

We explained the importance of EHP-5 in the QoL section, line 106-115: " Quality-of-life evaluation: The EHP-5 (Endometriosis Health Profile) is a short version of the EHP-30 (HRQoL: Health Related Quality of Life), a specific questionnaire about quality of life in cases of endometriosis. It has recently been validated in French [15]. The EHP-5 addresses the patient’s social and sexual life, daily life management of pain, impact on work, emotions, relationship and potential infertility with 11 questions. The answer to each questions involves five levels ranging in order of severity, as “never”: 0, “rarely”: 25, “sometimes”: 50, “often”: 75 and “always”: 100. The higher the score, the greater the effect on quality of life. The best score possible for each question was 0 and the worst score was 100. The EHP-5 is shown in Figure 1.”

Point 5).

Have other concurrent pathologies been identified at the gynecological level?

Response 5).

No other pathology has been identified so far for these patients who are regularly seen in consultation to adapt the treatment if necessary as we mentioned line 79-80: “No case of misdiagnosis or especially malignant tumor was revealed on histopathology analysis.

Point 6).

Some comparative discussions of the therapeutic options appear necessary

Response 6).

We added a sentence in the discussion to explain the necessity of holistic approach:

Lines 251-254: " Furthermore, a third of patients still needed level 3 painkiller and hormonal treatment after surgery, showing that management of endometriosis required a holistic approach in order to be effective alongside surgery (painkillers, hormonal treatments, diet and lifestyle rules) [1]. “

Reviewer 3 Report

The authors investigated in the present study post-operative quality of life in patients 2 years after minimally invasive surgery for pain and deep infiltrating endometriosis. The study is original and very interesting, given that long term outcomes are rarely presented in this topic. Its main limitation is the retrieval of patients from a retrospective, rather than prospective, list, which renders it prone to potential bias.

The authors have correctly identified potential limitations with the potential exception of the relatively small study size. While a sample size calculation was not predesigned, a power calculation based on the actual finidngs of this study would significantly benefit the results and help readers understand if the authors have succesfully identified differences among included groups.

Figures have extremely small captions and these should be either increased or data entered in a appropriate legend.

Given the rarity of the reported results, it would be nice to introduce a section in the discussion where the authors would recommend what should be investigated by future researchers.

Several outcomes are presented and the authors should elaborate on them rather than just comparing them to previous studies. ex they state that the rsik of recurrence was higher in the conserv group (this is known), however, they should also mention if the actual difference in the percentage is clinically meaningful and if yes, if this difference is actually accepted by patients that request a conservative approach.

Several results are duplicated in tables and the text. Given that the data exist in tables the authors should consider ommiting them from the text.

Author Response

Reviewer 3:

We would like to thank reviewer number 3 for his encouragements and pertinent comments regarding our article about QoL after minimally invasive surgery.

Answers to reviewer 3:

 Point 1).

 The authors investigated in the present study post-operative quality of life in patients 2 years after minimally invasive surgery for pain and deep infiltrating endometriosis. The study is original and very interesting, given that long term outcomes are rarely presented in this topic. Its main limitation is the retrieval of patients from a retrospective, rather than prospective, list, which renders it prone to potential bias.

Response 1).

Thank for this first and global remark about our study, in fact, it's a retrospective cohort study with all the known bias associated.

We insisted on this aspect: lines: 231-235: " This is the first study showing that robot laparoscopy has a tendency to a superiority, but these results must be balanced with the limitations of our study which was retrospective. Indeed, the groups were not comparable especially concerning the rate of hysterectomy, which was higher in RL and could be a major bias. »

- lines 256-262: "The limitations were that we used retrospective data, which involves known associated bias such as loss of follow-up and non-comparable groups. We also had a small effective, with no preliminary power calculation, which strongly limited our conclusions. Patients who had taken a laparotomy route were also excluded. Follow-up was limited to 2 years, with no long-term data. Also, complications about prolapsus and persistent urinary retention were not collected. Prospective evaluation with a larger sample size is necessary for future studies.”

Point 2).

 The authors have correctly identified potential limitations with the potential exception of the relatively small study size. While a sample size calculation was not predesigned, a power calculation based on the actual findings of this study would significantly benefit the results and help readers understand if the authors have successfully identified differences among included groups.

Response 2).

Once more, thank you for your accurate look regarding statistics in this article. We calculated the power of the study with our final for the main objective. We added a specific paragraph to explain the methods of calculation of the power: Lines 142-143:

With a two-sided significant level of alpha=0.05, the power of the study was 88.36% on the main objective.”

Point 3).

 Figures have extremely small captions, and these should be either increased or data entered in an appropriate legend.

Response 3).

I would like to thank you to help us to see the flaws in all aspects of this article. We changed all the figures with bigger captions and colors to help lectors for a better understanding.

Point 4).

 Given the rarity of the reported results, it would be nice to introduce a section in the discussion where the authors would recommend what should be investigated by futur researchers.

Response 4).

We would like to thank you again for your support and encouragements, which are precious for a searcher. We found those results interesting, but some bias can limit the interpretation. As you suggested, we added a new section after the discussion to explain new prospects and argue them. Lines: 268-274:

5. Prospects. Surgical treatment of endometriosis needs a specific strategy for each patient. Today, it becomes clear that every surgical treatment has issues (on a short, mid- and long-term basis). More data is necessary to compare the benefits of the robotic versus laparoscopic way, as well as the conservative and total surgeries for deep painful endometriosis: prospective evaluation with a bigger effective, a long-term follow up, fertility evaluation, the recurrence rate of pain and the quality of life is mandatory”.

Point 5).

 Several outcomes are presented, and the authors should elaborate on them rather than just comparing them to previous studies. ex they state that the risk of recurrence was higher in the conserv group (this is known), however, they should also mention if the actual difference in the percentage is clinically meaningful and if yes, if this difference is actually accepted by patients that request a conservative approach.

Response 5).

You are right, our results didn't reach the significant threshold but may be clinically significant.

We rephrased it lines: 179-180: " No difference was found for the recurrence rate of painful symptoms comparing CL and RL group [CL: 17.8% (5/28) and RL: 3.8% (1/26); p=.19]. “

Lines 246-248: “Nevertheless, the rate of recurrence is clinically meaningful in CL group and should be explained to patients before surgery”

Point 6).

 Several results are duplicated in tables and in the text. Considering data exist in tables, the authors should consider omitting them from the text.

Response 6).

Thank you for this comment, In fact, understanding of results is too dense to read with too much data and as you said which are repeated. All the results have been simplified. Only the main results are presented in the article and all others information can be found in table 1 and 2, that have been modified.

Reviewer 4 Report

Please see the attached word file.

Author Response

Reviewer 4:

Answers to reviewer 4:

First, we would like to thank you for the interest that you reach to our work, and your support and encouragements. We really appreciate it.

It wasn't easy to manage these retrospective data and make them appeared interesting despite of some structural bias.

It's maybe a new door opening on surgery in the angle of QoL, making surgeons rethink their surgery indications. Radicality and QoL needed to be assessed. All these comments helped us to better structure our article and make it easier to understand and to refine it.

Point 1). ABSTRACT

Line 27:“VAS” needs formal expression at its first appearance.

Response 1).

We corrected it to "Visual Analogic Scale" at its first appearance line 28-29:" Visual Analogic Scale (VAS) was also lower for dysmenorrhea (pre-op: 8 (7-9.75) vs 2-y post-op: 3 (2-5.25); p<.001) and dyspareunia (pre-op: 6 (3.1–8.9) vs 2-y post-op: 3 (0–6); p<.001)".

Line 31; “ConserV” misspelling.

Thank you, we read the article several times but it's clearly an error that have been corrected, line 32.

Point 2). INTRODUCTION

Line 57; “EPH” needs formal expression at its first appearance.

Response 2).

Like the others remarks that you made, it's clearly an inattention mistake. Another reviewer pointed this lack of explanation of the origin of the questionnaire name which is the Endometriosis Health Profile. We corrected it lines 21-22.

Point 3). RESULTS

- Line 124; “EVA” needs formal expression at its first appearance.

It's a mistake, EVA is the French acronym for VAS. We corrected it line 93-95:" All data were collected from medical records: patient characteristics (age, BMI, history of prior surgery for endometriosis or other, pre- and post-operative painkiller and hormonal treatment), pre- and post-operative pain evaluation (visual analogue scale: VAS for dysmenorrhea and dyspareunia, use of level 3 painkiller, type of surgery (RL or CL)"

- Line 130; “VAS” needs formal expression again at its first appearance in the body of manuscript apart from abstract.

We corrected our mistake and explained the acronym like you asked, line 135.

- Line 141; “LUS” needs formal expression at its first appearance.

We deleted it line 141, because it was a repetition of tables. But we corrected it, to Utero sacral Ligament (USL) line 175.

- Line 171-173; “Post∼p<0.01).” is redundant and is better to be deleted.

Thank you for this comment, it appears that the section of results, was too dense, with repeated data. Instead of removing all p value, we deleted some repeated data, with a more fluent result. Another reviewer pointed that problem of density of the results section and its opacity.

- Line 179; “RAL” is RL?

You are right, we are talking about robotic surgery group RL, RAL is again the French spelling of this same surgical technique. It’s clearly an omission, we are sorry of this lack of precision in the work proofreading. We corrected it line 176.

- Line 179-180; The duration unit needs to be indicated. Months?

We corrected it, the duration of analgesia was expressed in minutes and has been corrected lines 178-179.

- Line 180-181; P value is 0.19. The reviewer thinks the judgement that there was difference is invalid. Typo?

We corrected it, because you are right, p value doesn’t reach the significant threshold, we can not conclude to a difference between the groups. You can see the new formulation line 179-181:

" No difference was found for the recurrence rate of painful symptoms comparing CL and RL group (CL: 17.8% (5/28) and RL: 3.8% (1/26); p=.19)."

- Line 182-183; “An improvement∼ p<0.01).” is redundant, is better to be deleted.

Considering the first comment about redundance, we deleted some parts, to make the results section easier and more comfortable to read. Several reviewers pointed the too important density with repeated data in this section.

- Line 190-191; P value is 0.38. The reviewer thinks the judgement that there was difference is invalid. Typo?

Like in the precedent answer, you are right, and with the same argument, p doesn't reach the significant threshold. We changed our conclusion line 186-187 to "No difference was found for recurrence rate of pain between ConservS and TS groups (ConservS: 15.6% (5/32) and TS: 4.5% (1/22); p= 0.38)".

Point 4). DISCUSSION

Line 210- 213; Showing the results with the number is not necessary in discussion.

We decided to delete all the numbers in this section, because you are right, we already presented this in the results section.

New paragraph is line 206-208: " Endometriosis patients who undergone surgery in our center had a very altered quality-of-life, elevated VAS for dysmenorrhea and dyspareunia, indicating very painful symptoms after failure of all medical treatment".

Line 243-244; The sentence should be moved to material and methods.

It was more pertinent to put this important argument in the section material and methods, we follow your advice and replaced it to line 83-87 " We also evaluated the total resection of endometriosis (excision of all lesions) compared to conservative Surgery in this study. Conservative surgery was the targeted resection of endometriosis with a left residual endometriosis to minimise post-operative complications in case of frozen pelvis, rectal or urinary risk”.

Line 247; “RPM” needs formal expression at its first appearance.

Thank you again for your attention, RPM is the French translation to talk about Persistent Urinary Retention (UR). We changed it line 242.

Round 2

Reviewer 1 Report

The reviewer is satisfied with the revision.

Reviewer 4 Report

The reviewer is satisfied with the revision.